# Surgical Outcomes and Trends for Chronic Pancreatitis: An Observational Cohort Study from a High-Volume Centre

**DOI:** 10.3390/jcm11082105

**Published:** 2022-04-09

**Authors:** Poya Ghorbani, Rimon Dankha, Rosa Brisson, Melroy A. D’Souza, Johannes-Matthias Löhr, Ernesto Sparrelid, Miroslav Vujasinovic

**Affiliations:** 1Division of Surgery, Department of Clinical Science, Intervention and Technology, Karolinska Institutet, Karolinska University Hospital, 14186 Stockholm, Sweden; rimon.dankha@stud.ki.se (R.D.); rosa.brisson@regionstockholm.se (R.B.); melroy.dsouza@ki.se (M.A.D.); matthias.lohr@ki.se (J.-M.L.); ernesto.sparrelid@ki.se (E.S.); 2Department of Upper Abdominal Diseases, Karolinska University Hospital, 14186 Stockholm, Sweden; miroslav.vujasinovic@ki.se; 3Department of Medicine, Karolinska Institutet, 14186 Stockholm, Sweden

**Keywords:** chronic pancreatitis, duodenum-preserving pancreatic head resection, high-volume centre, pancreaticoduodenectomy, surgical treatment

## Abstract

Surgery for chronic pancreatitis (CP) is considered as a last resort treatment. The present study aims to determine the short- and medium-term outcomes of surgical treatment for CP with a comparison between duodenum-preserving pancreatic head resection (DPPHR) and pancreatoduodenectomy (PD). The trends in surgical procedures were also examined. This was a retrospective cohort study of patients who underwent surgery for CP between 2000 and 2019 at the Karolinska University Hospital. One hundred and sixty-two patients were included. Surgery performed included drainage procedures (*n* = 2), DPPHR (*n* = 35), resections (*n* = 114, of these PD in *n* = 65) and other procedures (*n* = 11). Morbidity occurred in 17%, and the 90-day mortality was 1%. Complete or partial pain relief was achieved in 65% of patients. No significant difference in morbidity was observed between the DPPHR and PD groups: 17% vs. 20% (*p* = 0.728). Pain relief did not differ between the groups (62% for DPPHR vs. 73% for PD, *p* = 0.142). The frequency of performed DPPHR decreased, whereas the rate of PD remained unaltered. Surgical treatment for CP is safe and effective. DPPHR and PD are comparable regarding post-operative morbidity and are equally effective in achieving pain relief. Trends over time revealed PD as more commonly performed compared to DPPHR.

## 1. Introduction

Current treatment strategies for chronic pancreatitis (CP) follow a multidisciplinary step-up approach with medical therapy, endoscopic intervention and, lastly, surgery [1]. The main indications for surgery are intractable pain and suspicion of malignancy [2]. Furthermore, surgery has to be considered in patients with duodenal obstruction and pancreatic or common bile duct stenosis resistant to endoscopic measures [3]. Historically, surgery has been the last resort for CP patients, due to its technically demanding and invasive nature, with a substantial postoperative morbidity reported to be as high as 53.3% and a non-negligible overall mortality ranging from 1% to 19% [4]. Nevertheless, advancements in modern surgical techniques, the centralisation of care and improved perioperative management have resulted in mortality rates <5% and excellent long-term outcomes after tailored surgery [4,5]. Additionally, there is convincing evidence that earlier surgical intervention for CP is beneficial with regard to overall pain relief, quality of life (QoL) and endocrine function [6,7,8].

The surgical armamentarium available for the treatment of CP can be divided into three categories: pancreatic ductal drainage (e.g., Partington–Rochelle procedure) [9]; different types of pancreatic resections (e.g., pancreaticoduodenectomy (PD), distal pancreatectomy (DP) and total pancreatectomy (TP) [10,11]; and a combination of resection and drainage procedures, commonly referred to as duodenum-preserving pancreatic head resection (DPPHR), such as the Beger [12] and Frey procedures [13]. Although the choice of procedure mainly depends on the anatomical abnormalities and morphological changes of the pancreas [14], there is uncertainty concerning the optimal surgical approach (especially with regard to DPPHR vs. PD) for CP patients with chronic pain originating from an inflammatory mass of the pancreatic head [15]. Several single centre trials have compared the various techniques of DPPHR with PD, suggesting that DPPHR should be the procedure of choice due to favourable short-term outcomes [16,17,18,19]. However, a recent multicentre randomised trial could not confirm the superiority of DPPHR over PD [20]. Moreover, this topic was addressed by two systematic reviews and meta-analyses. Zhao et al. [21] concluded that DPPHR is a more beneficial surgical strategy for CP owing to improved short- and long-term outcomes. Contrarily, Gurusamy et al. [22] established that there were no differences between DPPHR and PD with regard to similar endpoints.

The Karolinska University Hospital is a high-volume tertiary referral centre for pancreatic diseases, responsible for the medical, endoscopic and surgical treatment of CP. The primary aim of this study is to investigate the short- and medium-term outcomes of surgery for CP at this centre, with an emphasis on the comparison between DPPHR and PD. A secondary aim is to evaluate the trends in surgical procedures over time.

## 2. Methods

This study was performed in accordance with the Strengthening the Reporting of Observational Studies in Epidemiology (STROBE) guidelines [23], and approved by the Regional Ethics Committee for human studies, Stockholm, Sweden (reference number 2016/1571-31).

### 2.1. Study Design and Study Population

This was a retrospective observational cohort study on prospectively collected data of consecutive adult patients (age ≥ 18 years) who underwent elective surgical intervention for CP between 1 January 2000 and 31 December 2019 at the Karolinska University Hospital in Stockholm, Sweden. Patients who had no evidence of CP on the final pathology report were excluded.

Patients were selected from a retrospectively maintained database of CP patients, and their medical records were analysed. The diagnosis of CP was established, based on the findings of clinical history, clinical examination and imaging according to the M-ANNHEIM classification of CP [24]. Patient characteristics, postoperative outcomes and the impact of surgery on exocrine and endocrine function, as well as pain relief at a 6-month follow-up, were evaluated.

### 2.2. Indications for Surgery and the Selection of Surgical Procedure

The treatment strategy for CP at our centre was similar to the standard step-up approach with initial medical therapy, followed by endoscopic intervention and lastly surgery. Hence, surgical treatment was reserved for patients with intractable abdominal pain after failure of medical and/or endoscopic intervention. Other indications comprised suspicion of malignancy, locoregional complications, such as biliary and duodenal obstruction due to an inflammatory mass of the pancreatic head, and symptomatic pseudocysts. The decision for surgical treatment was taken following multidisciplinary meetings with surgeons, radiologists, endoscopists and pancreatologists. The type of surgery performed was decided, based on the specific pancreatic morphological features.

### 2.3. Preoperative Data

The following parameters were collected: sex; age; aetiology of CP; history of smoking; time between CP diagnosis and surgery (months); number of endoscopic retrograde cholangiopancreatography (ERCP) procedures performed prior to surgery; presence of calcification on radiology; indication for surgery; presence of pancreatic exocrine insufficiency (PEI); dosage of pancreatic enzyme replacement therapy (PERT); body mass index (BMI); stool consistency; diabetes mellitus (DM); pain treatment and pain intensity.

### 2.4. Short-Term Postoperative Outcomes

The following parameters were collected: intraoperative blood loss (millilitres); operative time (minutes); hospital length of stay (LOS, days); LOS at the high-dependency unit (HDU, i.e., level of care intermediate between intensive care and general ward); intensive care unit (ICU) admission and ICU LOS; duration of epidural anaesthesia usage (days); duration of patient-controlled analgesia usage (days); postoperative morbidity; delayed gastric emptying (DGE); postoperative pancreatic fistula (POPF); Postpancreatectomy haemorrhage (PPH) and bile leakage.

### 2.5. Medium-Term Outcomes

The outpatient follow-up at our centre was scheduled at 3 and 6 months postoperatively. Thus, data at the 6-month follow-up were retrieved and evaluated for medium-term outcomes, including PEI, BMI, stool consistency, DM treatment, pain medication and pain intensity. The 6-month cut-off ensured an adequate time for patients to recover from surgery and adapt to their insufficiencies, thus improving the homogeneity of the cohort, and also reduced the amount of missing data.

### 2.6. Trends in the Surgical Procedures

For the purposes of the analysis of trends in surgical procedures, the study period was arbitrarily sub-classified into two periods (2000 to 2010 and 2011 to 2019). The surgical procedures evaluated were DPPHR, PD, DP and TP.

### 2.7. Definitions

PEI was defined as faecal elastase-1 values <200 µg/g or steatorrhea [25]. Stool consistency was estimated using the Bristol Stool Form Scale [26] and divided into three groups (type 1–2 = constipation, type 3–5 = normal and type 6–7 = diarrhoea). Patient’s subjective self-reported pain intensity using the visual analogue scale (VAS) [27] in the last outpatient visit prior to surgery was obtained and divided into three pain levels (none = VAS 0, mild/moderate = VAS 1–5 and severe pain = VAS 6–10) [28]. Pain relief at follow-up was defined as the sum of the complete alleviation of symptoms and partial relief in pain level (i.e., reduction from severe pain to moderate/mild pain). Endocrine insufficiency was defined as the presence of DM, which was diagnosed according to the World Health Organization diagnostic criteria for DM [29]. Opioid doses were standardised to morphine milligram equivalents using the conversion guide provided by Pfizer AB [30]. Surgical morbidity was classified according to the Clavien–Dindo classification system [31]: only major morbidity defined as grade 3a–5 was deemed clinically relevant. POPF was defined and graded according to the International Study Group of Pancreatic Fistula [32]. Bile leakage was categorised according to the International Study Group of Liver Surgery [33]. Furthermore, PPH [34] and DGE [35] were classified according to the International Study Group of Pancreatic Surgery: only grade B and C complications were considered to be of clinical relevance.

### 2.8. Statistical Analyses

Descriptive results for numerical variables were presented as the means with standard deviation (±SD) in normally distributed data and as the median with interquartile range (IQR) in non-normally distributed data. Frequency distributions and percentages were used to summarise the categorical variables. Comparisons of numerical data were performed using the independent *t*-test or Mann–Whitney U test, depending on the distribution of the data. Categorical variables were compared using Pearson’s chi-squared test or Fisher’s exact test, when appropriate. Comparisons of pre- and postoperative data were performed using McNemar’s test and the Wilcoxon signed-rank test on nominal and non-parametric data, respectively. A normality was tested using the Shapiro–Wilk test. Statistical significance was assumed for a 2-sided *p*-value < 0.05. Statistical analysis was performed using the software SPSS Statistical Package for the Social Sciences (Version 23.0.0; SPSS, Inc., Chicago, IL, USA).

## 3. Results

### 3.1. Final Cohort and Type of Surgical Procedure

There were 162 patients included in the study. Pancreatic resection was the most common procedure (70%), the majority being PD, 65 out of 162 (40%). A total of 35 patients (22%) underwent DPPHR. Of these, 23 patients (14%) underwent the Beger procedure and 3 patients (2%), the Frey procedure. Surgical drainage procedures were performed on 2 patients (1%), according to Partington–Rochelle. See Figure 1 for patient inclusion and surgical procedures.

### 3.2. Patient Characteristics

Clinical characteristics are presented in Table 1. There were 63 females (39%) and 99 males (61%). The mean age was 54 (±14) years at the time of surgery. The most common aetiology of CP was excessive alcohol consumption (44%). The majority of patients were smokers (70%), with 38% reporting lifetime cigarette consumption between 21 and 40 pack years. The median time between the diagnosis and surgery was 14 (5–34) months. The most common indication for surgery was a primary suspicion of malignancy, in addition to CP complications in 101 patients (62%). Of these, only 10 patients (9.9%) had confirmed pancreatic adenocarcinoma with underlying CP in the final pathology report. The principal indication for patients who underwent surgery strictly due to CP complications was intractable pain in 33 out of 61 patients (54%). Pancreatic abscesses, pseudocysts and disease progression were other less common indications for surgery.

Patients in the DPPHR group were younger (mean 48 [±14] vs. 58 [±12] years, *p* < 0.001) than the PD group, and underwent a greater number of endoscopic interventions (ERCP) prior to surgery (*p* = 0.005). The indications for surgery were different between the 2 groups, in which CP complications accounted for 74% of all indications in the DPPHR group compared to 18% in the PD group (*p* < 0.001). There was also a higher rate of radiologically confirmed pancreatic calcification in patients who underwent DPPHR, compared to PD, 71% vs. 42%, (*p* = 0.006).

### 3.3. Short-Term Outcomes of Surgery

The short-term outcomes are summarised in Table 2. In total, the median postoperative hospital LOS was 14 (11–18) days. A total of 21 patients (13%) were admitted to the ICU. Major morbidity (Clavien–Dindo 3a–5) occurred in 27 patients (17%). Additionally, 54 patients (33%) suffered from morbidity related to DGE, POPF, PPH or bile leakage, among which PPH was the most common (*n* = 23, 44%). The 90-day mortality was 1% (*n* = 2).

There was no difference in hospital or ICU LOS between the DPPHR and PD groups. Regarding morbidity (Clavien–Dindo grades 3a–5 and DGE, POPF, PPH, bile leakage grades B/C) and mortality (in-hospital and 90-day), the two groups did not differ. Patient-controlled analgesia was required more frequently in the DPPHR group (*p* < 0.001) compared to PD, in adjunct to epidural analgesia, which was used in 97% of all cases.

### 3.4. Medium-Term Outcomes of Surgery

The medium-term outcomes and comparison of non-surgical pre- and postoperative data for DPPHR and PD are shown in Table 3 and Table 4. The comparisons of non-surgical data for the entire cohort are presented in Appendix A: non-surgical data before and after surgery for the entire cohort.

### 3.5. Pancreatic Exocrine Insufficiency

For the entire cohort, PEI was noted in 70 patients (45%) prior to surgery. A total of 72 patients (46%) developed new-onset PEI, resulting in a total of 142 patients (90%) with PEI at the 6-month postoperative evaluation.

There was an increase in the number of PEI cases postoperatively in both the DPPHR and PD groups, compared to preoperatively (*p* < 0.001, respectively). New-onset PEI developed in 12 patients (34%) in the DPPHR group compared to 37 patients (59%) who underwent PD, which differed significantly between the 2 groups (*p* = 0.020).

### 3.6. Endocrine Insufficiency

In the total cohort, 30% (48 out of 162) had DM prior to surgery. A total of 29 of 105 patients (28%) (9 missing cases) developed new-onset DM. DM was thus present in 77 cases (50%) at follow-up.

The proportion of patients with DM increased postoperatively, although not significantly, from 29% to 44% (*p* = 0.063) in the DPPHR group. A similar relation was observed after PD, from 38% to 48%, (*p* = 0.031), this time significant. There was, however, no difference in the proportion of new-onset DM nor total proportion of DM between the two procedures (*p* = 0.515 and *p* = 0.748, respectively).

### 3.7. Pain Management

For the entire cohort, 69 out of 121 patients (57%, 41 missing cases) reported having abdominal pain of varying severity before surgery. Complete or partial pain relief was achieved in 45 out of the 69 patients (65%). In total, 42 out of 121 patients (35%, 41 missing cases) experienced pain after surgery, thus resulting in a reduction in overall pain when comparing pain before and after surgery (69 patients vs. 42 patients, *p* < 0.001).

Opioid consumption was higher at 6-month follow-up among patients who underwent DPPHR, compared to PD (54% vs. 25%, *p* = 0.009). Additionally, the DPPHR group had higher pain levels (mild/moderate and severe pain) at follow-up compared with the PD group (46% vs. 26%, *p* = 0.044). For the DPPHR group, there was no difference between pre- and postoperative opioid consumption. In the PD group, however, there was a tendency toward less opioid consumption at postoperative follow-up compared to preoperatively (*n* = 14, 25% vs. *n* = 22, 39%, *p* = 0.077).

Regarding pain relief, 21 patients (81%) in the DPPHR group experienced abdominal pain (mild/moderate or severe pain) prior to surgery, and, of these, complete or partial pain relief was achieved in 13 patients (62%) (*p* = 0.021). For the PD group, 22 patients (44%) experienced abdominal pain prior to surgery, and, of these, complete or partial pain relief was noted in 16 patients (73%) (*p* = 0.022). Moreover, the overall difference between subjectively reported pain levels before and after DPPHR and PD interventions was significantly lower (*p* = 0.012 and *p* = 0.003, respectively). However, the differences in pain relief between the two groups was not statistically significant (*p* = 0.142).

### 3.8. Trends in Surgical Procedures

Figure 2 provides an overview of the trends in the type of surgical procedures (only DPPHR and resections included) during the study period. The most used surgical procedure was PD, followed by DPPHR, DP and TP. There was a difference in the proportion of the surgical procedures between the time periods 2000–2010 and 2011–2019 (*p* < 0.001). The frequency of DPPHR decreased from 41% during the first decade to 5% in the following years. In fact, no patient underwent DPPHR in the last seven years of the study period. Conversely, the rates of DP and TP increased from 8% to 31% and from 4% to 14%, respectively, during the same time period. The rate of PD remained largely unaltered throughout the years.

## 4. Discussion

To evaluate the safety and efficacy of surgical treatment for CP, we conducted a retrospective cohort study of 162 CP patients undergoing surgery. The results suggest that surgical treatment for CP can be carried out safely with acceptable pain relief. Furthermore, both DPPHR and PD were found to be equally safe. In addition, the two procedures did not differ in terms of pain relief or endocrine insufficiency, although the DPPHR group experienced higher pain levels and opioid consumption at the 6-month follow-up. Trends over time revealed pancreatic resections (PD, DP and TP) as more commonly performed at our centre compared to DPPHR.

The current study revealed low overall rates of major morbidity (17%) and 90-day mortality (1%). The results are in line with pervious series (19–28% and 1–4%, respectively) [5,36,37]. The median hospital LOS in these studies ranged between 9–15 days, which is also comparable to the present study (14 days). The demonstrated morbidity, 90-day mortality rate <5% and relatively short postoperative hospital stay underlines the impact of centralisation of care to high-volume centres specialised in pancreatic surgery.

The incidence of endocrine and exocrine insufficiency is common in long-term follow-up series, ranging between 0–34% and 27–37%, respectively [37,38,39]. Similar to these series, new onset DM occurred in 28% of our patients. New onset PEI, however, developed in 46% of our patients. It is possible that this difference is related to how PEI was measured. Most studies examine PEI by objective methods, such as faecal elastase-1 test, whereas PEI, in our study, was defined as presence of steatorrhea or reduced faecal elastase-1 values. Furthermore, it has been suggested that the decline in endocrine and exocrine function is independent of whether a conservative approach or surgical treatment is employed [38]. In the long run, most CP patients will develop pancreatic dysfunction [38].

The main criterion of success in the surgical management for CP is the achievement of pain relief [40,41,42]. In the present study, complete or partial pain relief was achieved in 65% of the total population over a follow-up period of 6 months, regardless of indication for surgery. These findings are not as promising when compared to previous studies reporting complete pain relief in more than 80% of patients [37,40,41,43]. This disparity could have several explanations. Firstly, the course of pain in CP can be variable. In the initial stages, the pain is intermittent, and as the disease progresses, the pain becomes more constant [3]. When analysing pain data, patients were classified as pain-free if there was no report of abdominal pain in the last outpatient clinic visit prior to surgery, irrespective of symptomatology within the last 6 months. This would naturally exclude patients with intermittent abdominal pain. Secondly, the main indication for surgery in our study group was the suspicion of malignancy in addition to CP complications. Therefore, one could make the inference that greater pain relief can be achieved when the predominant preoperative indication is intractable pain, instead. Thirdly, it has been suggested that a shorter postoperative follow-up period (less than 5 years) is associated with a lower rate of pain relief [37].

Our study is unusual in its short median duration (14 months) between CP diagnosis and surgical intervention. Several studies reported a median duration of 40 months or more [5,37]. This is likely due to the higher proportion of CP patients in our cohort with a suspicion of malignancy (62%), albeit in line with what has previously been reported [44], and our department’s stringency regarding this parameter. Furthermore, several studies suggested that the timing of surgery is an important factor in improving clinical outcomes following surgery [14,45]. Moreover, the recently published ESCAPE trial concluded that early surgery was associated with lower pain scores, compared to the endoscopy-first approach (6).

In our comparison of DPPHR with PD, we found that both procedures were equally safe concerning morbidity (17% vs. 20%) and 90-day mortality (0% vs. 3%). These findings are in concordance with the previously published data [17,39,46]. However, Aspelund et al. [16,47] observed a significantly higher morbidity among patients who underwent PD (40%), compared to the DPPHR group (25%) and the Partington–Rochelle group (16%).

In the current study, patients who underwent DPPHR procedures had continued pain and higher amounts of opioid consumption at follow-up, compared to the PD group. However, the two groups did not differ significantly regarding complete or partial pain relief (62% vs. 73%, *p* = 0.142). Thus, our results provide further evidence that DPPHR and PD interventions are equally effective concerning pain relief. Nevertheless, regarding the discrepancy in postoperative pain levels, one should consider that patients who underwent DPPHR had a higher number of ERCP procedures prior to surgery, a significantly higher rate of pancreatic calcifications, higher opioid consumption, and a longer duration of symptoms. Therefore, it is reasonable that these preoperative characteristics might reflect a more advanced disease in this group, which could explain the slightly higher pain levels observed at follow-up. Concerning the difference in opioid consumption between the groups, it should be noted that there was a significantly higher proportion of patients in the DPPHR group who required opioids prior to surgery, 56% vs. 39%. More importantly, we found no significant difference in opioid consumption between preoperative and postoperative values in either group.

The present series found a significant shift in surgical procedures for CP during the study period. The trends in pancreatic surgery at our centre were nearly identical to the observations reported by Wittel et al. between 1994 and 2012 [48]. The authors suggested that the reduction in the DPPHR rate was largely due to increased malignant indications for pancreatic surgery as opposed to benign diagnoses, such as CP. Similarly, we observed that CP patients were less likely to undergo surgery solely due to abdominal pain in the latter part of the study period, compared to early on (data not shown). All the while, the main indication for surgical intervention continued to be a suspicion of malignancy, which, for all intents and purposes, rendered the various DPPHR procedures ineffective. Furthermore, an international survey among pancreatic surgeons found that US surgeons tend to favour PD, with 25 out of 59 surgeons reporting to never having performed a single DPPHR [49]. In contrast, DPPHR was the preferred choice of surgical therapy for CP in Germany. Keck et al. [50] suggested that the disparity in surgical approach between American and German pancreatic centres could be explained by the pathomorphology of the pancreas among the CP patients. All things considered, the choice between DPPHR and PD could simply reflect institutional preferences and local expertise, which could also be the case at our centre.

The results of this study should be interpreted considering several limitations. Firstly, there is the inherent limitation of the retrospective nature of data collection and analysis in an historical cohort (such as missing data and temporal changes). Secondly, even though those with primary suspicion of malignancy as an indication for surgery also had CP complications, the two treatment groups (DPPHR vs. PD) were not entirely comparable in terms of surgical indications and pancreatic morphology. Thirdly, our 6-month follow-up period is relatively short compared to other similar studies. Although there is high-quality literature examining this field, the main strength of our study is its status as the first in-depth analysis of a Scandinavian population. Additionally, a relatively large number of patients from a single centre were included in the final analysis. Thus, detailed surgical and non-surgical results could be obtained. Finally, the long study period provided the opportunity to study the trends in pancreatic surgery for CP patients over time.

## 5. Conclusions

Surgical treatment for chronic pancreatitis is safe and effective when performed at a high-volume centre. In properly selected patients, surgical intervention is associated with acceptable low perioperative morbidity and mortality and provides adequate pain relief. Furthermore, DPPHR and PD procedures seemed to be equally effective with regard to morbidity, mortality, pain relief and endocrine preservation. These findings must be carefully interpreted considering the limited sample sizes and the retrospective nature of the study making new prospective studies warranted. Trends over time revealed pancreatic resection as more commonly performed compared to DPPHR, which may be due to increased malignant indications for pancreatic surgery.

## Figures and Tables

**Figure 1 jcm-11-02105-f001:**
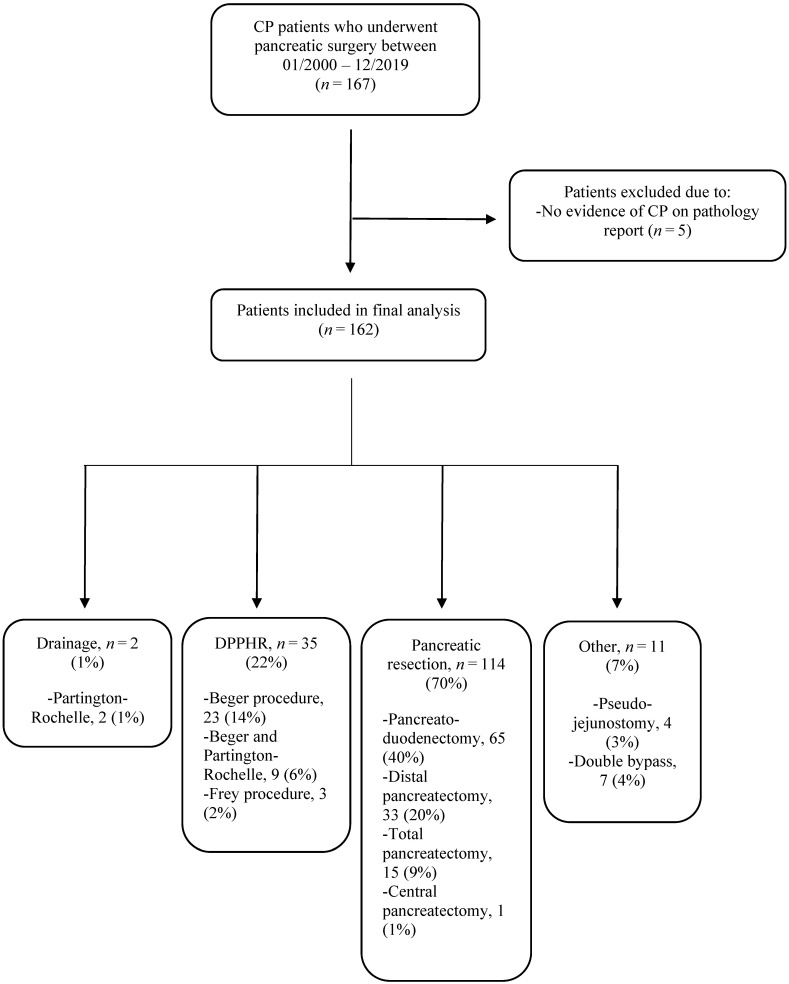
Patient inclusion and surgical procedures. CP: Chronic pancreatitis; DPPHR: Duodenum-preserving pancreatic head resection.

**Figure 2 jcm-11-02105-f002:**
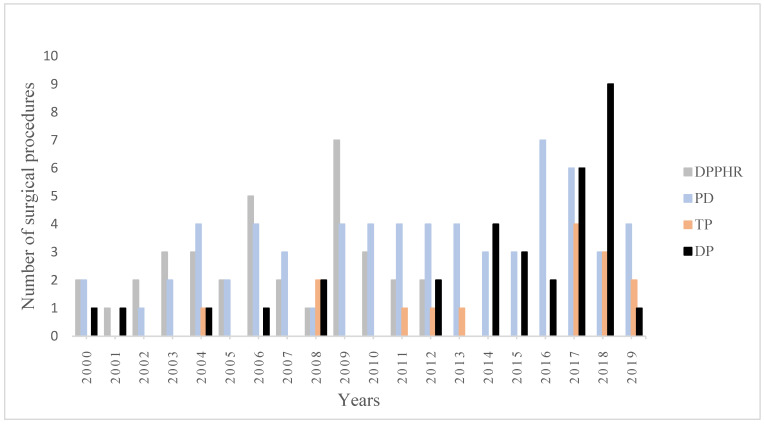
Trends over time in the surgical procedures for chronic pancreatitis at Karolinska University Hospital. DPPHR: duodenum-preserving pancreatic head resection; PD: pancreatoduodenectomy; TP: total pancreatectomy; and DP: distal pancreatectomy.

**Table 1 jcm-11-02105-t001:** Patient characteristics in the total population among duodenum-preserving pancreatic head resection (DPPHR) and pancreaticoduodenectomy (PD).

	Total *n* = 162	DPPHR *n* = 35	PD *n* = 65	*p*-Value ^a,b^	Missing Cases (*n*) Total/DPPHR/PD
Gender, *n* (%)				0.353	1/0/0
Female	63 (39)	14 (40)	20 (31)		
Male	99 (61)	21 (60)	45 (69)		
Age in years, mean (±SD)	54 (14)	48 (14)	58 (12)	**<0.001**	1/0/1
Aetiology of CP, *n* (%)				**0.025**	13/0/7
Alcohol	65 (44)	21 (60)	27 (47)		
Nicotine	26 (17)	4 (11)	12 (21)		
Hereditary	8 (5)	4 (11)	0 (0)		
Autoimmune	18 (12)	0 (0)	6 (10)		
Obstructive	25 (17)	5 (14)	9 (16)		
Miscellaneous	7 (5)	1 (3)	4 (7)		
Smoking history in pack years, *n* (%)				0.365	24/3/8
0	41 (30)	10 (31)	17 (30)		
1–20	33 (24)	10 (31)	11 (19)		
21–40	53 (38)	10 (31)	23 (40)		
41–60	9 (6)	2 (6)	4 (7)		
61–80	2 (1)	0 (0)	2 (4)		
Time between CP diagnosis and surgery in months, median (IQR) ^c^	14 (5–34)	15 (8–31)	8 (3–21)	0.067	40/1/22
Number of ERCP prior to surgery, median (IQR)	1 (0–2)	2 (0–3)	0 (0–1)	**0.005**	12/5/4
Pancreatic calcification at CP diagnosis, *n* (%)	68 (42)	24 (71)	27 (42)	**0.006**	2/1/0
Indications for surgery, *n* (%)				**<0.001**	0/0/0
CP complications Suspicion of malignancy	61 (38) 101 (62)	26 (74) 9 (26)	12 (18) 53 (82)		
Histologically verified malignancy, *n* (%)	10 (6)	0 (0)	3 (8)	0.550	3/0/1

^a^ *p*-values were calculated using the Mann–Whitney U test for non-normally distributed variables and ordinal data, independent *t*-test for normally distributed variables. Chi-squared test and Fisher’s exact test were used for categorical variables. ^b^ Bold values denote statistical significance (*p* < 0.05). CP: chronic pancreatitis; IQR: interquartile range; ERCP: endoscopic retrograde cholangiopancreatography; DPPHR: duodenum-preserving pancreatic head resection; and PD: pancreaticoduodenectomy. ^c^ Patients with a suspicion of malignancy without prior CP diagnosis were not accounted for.

**Table 2 jcm-11-02105-t002:** Short-term postoperative outcomes.

	Total *n * = 162	DPPHR *n* = 35	PD *n* = 65	*p*-Value ^a,b^	Missing Cases (*n*) Total/DPPHR/PD
LOS in days, median (IQR)	14 (11–18)	16 (11–18)	15 (11–22)	0.772	0/0/0
LOS-ICU in days, median (IQR)	0 (0–0)	0 (0–0)	0 (0–0)	0.500	0/0/0
LOS-HDU in days, median (IQR)	3 (1–5)	2 (0–4)	4 (2–6)	**<0.001**	0/0/0
DGE, *n* (%)					0/0/0
Grades B–C	17 (10)	3 (9)	10 (15)	0.534	
Bile leakage, *n* (%)					0/0/0
Grades B–C	5 (3)	1 (3)	4 (6)	0.655	
PPH, *n* (%)					0/0/0
Grades B–C	23 (14)	6 (17)	11 (17)	0.978	
POPF, *n* (%)					0/0/0
Grades B–C	9 (6)	1 (3)	3 (5)	1.000	
In-hospital mortality, *n* (%)	1 (1)	0 (0)	1 (2)	1.000	0/0/0
90-day mortality, *n* (%)	2 (1)	0 (0)	2 (3)	0.540	0/0/0
Clavien–Dindo, *n* (%)				0.728	0/0/0
Grades 3a–5	27 (17)	6 (17)	13 (20)		
EA postoperatively, *n* (%)	155 (97)	35 (100)	61 (95)	0.550	2/0/1
EA use in days, median (IQR)	6 (6–8)	8 (6–8)	6 (6–9)	0.392	0/0/0
PCA treatment postoperatively, *n* (%)	39 (24)	17 (49)	11 (17)	**<0.001**	0/0/0
PCA use in days, median (IQR)	0 (0–0)	0 (0–7)	0 (0–0)	**<0.001**	0/0/0
Oral opioids postoperatively, *n* (%)	152 (94)	30 (86)	60 (92)	0.313	0/0/0
Operative time in min, mean (±SD)	343 (122)	257 (77)	395 (107)	**0.008**	79/30/27
Perioperative blood loss in ml, median (IQR)	500 (200–1000)	700 (413–1038)	625 (250–1350)	0.464	35/19/9

^a^ *p*-values were calculated using the Mann–Whitney U test for non-normally distributed variables and ordinal data, independent *t*-test for normally distributed variables. Chi-squared test and Fisher’s exact test were used for categorical variables. ^b^ Bold values denote statistical significance (*p* < 0.05). LOS: length of stay; LOS-ICU: length of stay at the intensive care unit; LOS-HDU: length of stay at the high-dependency unit; DGE: delayed gastric emptying; PPH: postpancreatectomy haemorrhage; POPF: postoperative pancreatic fistula; EA: epidural anaesthesia; PCA: patient-controlled analgesia; IQR: interquartile range; DPPHR: duodenum-preserving pancreatic head resection; and PD: pancreaticoduodenectomy.

**Table 3 jcm-11-02105-t003:** Medium-term outcomes at the 6-month follow-up.

	Total *n* = 162	DPPHR *n* = 35	PD *n* = 65	*p*-Value ^a,b^	Missing Cases (*n*) Total/DPPHR/PD
PEI, *n* (%)	142 (90)	33 (94)	60 (95)	1.000	5/0/2
PERT dosage in lipase units, median (IQR)	150,000 (84,000–225,000)	168,000 (84,000–225,000)	150,000 (102,000–225,000)	0.769	24/6/12
BMI in kg/m^2^ median (IQR) and mean (± SD)	22.7 (19.4–25.7)	22.0 (3.2)	22.4 (4.5)	0.769	48/14/19
Stool consistency, *n* (%)				0.387	26/5/11
Constipation	9 (7)	1 (3)	0 (0)		
Normal	98 (72)	24 (80)	42 (78)		
Diarrhoea	29 (21)	5 (17)	12 (22)		
DM, *n* (%)	77 (50)	15 (44)	29 (48)	0.748	9/1/4
Treatment of DM, *n* (%)				0.581	10/1/4
Peroral	8 (5)	1 (3)	2 (3)		
Insulin dependent	56 (37)	13 (38)	21 (34)		
Peroral and insulin	12 (8)	1 (3)	6 (10)		
Pain treatment, *n* (%)					
Acetaminophen	61 (44)	15 (54)	21 (37)	0.142	24/7/8
NSAID	9 (6)	3 (11)	4 (7)	0.679	24/7/8
Opioid	43 (31)	15 (54)	14 (25)	**0.009**	25/7/9
Opioid dosage in mg, median (IQR)	0 (0–10)	5 (0–36)	0 (0–0)	**0.005**	30/9/12
Pain intensity, *n* (%)				**0.044**	41/7/14
No pain	85 (66.9)	15 (54)	38 (74)		
Mild/moderate pain	30 (23.6)	9 (32)	11 (22)		
Severe pain	12 (9.4)	4 (14)	2 (4)		

^a^ *p*-values were calculated using the Mann–Whitney U test for non-normally distributed variables and ordinal data. Chi-squared test and Fisher’s exact test were used for categorical variables. ^b^ Bold values denote statistical significance (*p* < 0.05). PEI: pancreatic exocrine insufficiency; PERT: pancreatic enzyme replacement therapy; BMI: body mass index; DM: diabetes mellitus; IQR: interquartile range; SD: standard deviation; NSAID: non-steroidal anti-inflammatory drug; DPPHR: duodenum-preserving pancreatic head resection; and PD: pancreaticoduodenectomy.

**Table 4 jcm-11-02105-t004:** Non-surgical data before and after duodenum-preserving pancreatic head resection (DPPHR) and pancreaticoduodenectomy (PD).

	Prior to		6 Months After		*p*-Value ^a,b,c^		Missing Cases (*n*)
	DPPHR *n* = 35	PD *n* = 65	DPPHR *n* = 35	PD *n* = 65	DPPHR	PD	DPPHR/PD
PEI, *n* (%)	21 (60)	23 (36)	33 (94)	60 (95)	**<0.001**	**<0.001**	0/2
PERT dosage in lipase units, median (IQR)	84,000 (0–150,000)	0 (0–117,000)	168,000 (84,000–225,000)	150,000 (102,000–225,000)	**0.001**	**<0.001**	6/12
BMI in kg/m^2^ median (IQR)	22.2 (20.4–24.7)	23.6 (21.6–28.3)	21.8 (19.2–24.6)	22.4 (19.0–24.6)	0.520	**<0.001**	14/19
Stool consistency, *n* (%)					0.739	1.000	10/20
Constipation	2 (8)	3 (7)	0 (0)	0 (0)			
Normal	17 (68)	28 (62)	20 (80)	34 (76)			
Diarrhoea	6 (24)	14 (31)	5 (20)	11 (24)			
DM, *n* (%)	10 (29)	23 (38)	15 (44)	29 (48)	0.063	**0.031**	1/4
Pain treatment, *n* (%)							
Acetaminophen	13 (48)	22 (39)	15 (56)	21 (37)	0.625	1.000	8/8
NSAID	5 (18)	5 (9)	3 (11)	5 (7)	0.687	1.000	8/8
Opioid	15 (56)	22 (39)	14 (52)	14 (25)	1.000	0.077	8/9
Opioid dosage in mg, median (IQR)	10 (0–70)	0 (0–23.5)	0 (0–38)	0 (0–0)	0.330	**0.027**	10/12
Pain intensity, *n* (%)					**0.012**	**0.003**	9/16
No pain	5 (19)	27 (55)	13 (50)	36 (74)			
Mild/moderate pain	7 (27)	13 (26)	9 (35)	11 (22)			
Severe pain	14 (54)	9 (18)	4 (15)	2 (4)			

^a^ Comparison between prior to surgery and 6 months after surgery in the DPPHR and PD groups, respectively. ^b^
*p*-values were calculated using McNemar’s test for categorical variables and Wilcoxon signed-rank test for non-normally distributed variables and ordinal data. ^c^ Bold values denote statistical significance (*p* < 0.05). PEI: pancreatic exocrine insufficiency; PERT: pancreatic enzyme replacement therapy; IQR: interquartile range; BMI: body mass index; DM: diabetes mellitus; NSAID: non-steroidal anti-inflammatory drug; DPPHR: duodenum-preserving pancreatic head resection; and PD: pancreaticoduodenectomy.

## Data Availability

The data collected and analysed during the current study are available from the corresponding author on reasonable request.

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
