# Peer review of "Surgical Outcomes and Trends for Chronic Pancreatitis: An Observational Cohort Study from a High-Volume Centre"

_jcm, 2022, doi:10.3390/jcm11082105_

Round 1
Reviewer 1 Report
This retrospective cohort study by Ghorbani et al describes surgical outcomes for pancreatic surgery secondary to chronic pancreatitis. There are two aims of this study, the first to delineate their current practice trends, and second, to compare DPPHR and PD in terms of outcomes. Overall, the researchers found that their outcomes were favorable and there was a decrease in utilization of DPPHR at their institution. This article addresses the important problem of chronic pancreatitis, a disease process that is not well understood.
Methods:
- The authors describe this as a retrospective observational cohort study. Could they clarify if the data was collected prospectively and analyzed retrospectively? Or if the data was abstracted and analyzed retrospectively?
- Patients were included from a 20-year period (2000-2019). Were there any major changes in care over time that could have affected outcomes, such as changes in diagnostic modalities, patients’ selection, pain control protocols, etc? If so, how did the authors control for temporal changes?
- High dependency unit is not a universal term. Can the authors clarify what a HDU is?
- For the trends in surgical procedures, the time cohort was split into two periods. Was this an arbitrary date cutoff or chosen because there was a change in practice? Furthermore, it appears that DPPHR use significantly decreased in the latter portion of the time period. Based on the presented data, the most common indication for this procedure was concern for malignancy. Has there been any advancements in diagnosis of malignancy that has resulted in reduced utilization of DPPHR and how will you account for this when comparing outcomes?
- Did the authors consider performing a multivariate analysis to adjust for confounders such as year, surgeon, age, gender, etc.?
- How was missingness accounted for in the statistical analysis? There were a significant number of patients for whom data was missing for some variables and outcomes. The authors should discuss this in their limitations section
- Did the authors perform a power analysis for post-op morbidity, mortality, and other important outcomes? The sample size is small despite the institution being a high-volume center for these procedures
Results
- It seems from the results that the role of DPPHR and PD differ based on the suspicion of malignancy. This suspicion of malignancy is a confounding factor. Therefore, although a patient may have CP, the true reason to pick one surgery versus another could be the fact that there is a risk of malignancy, not the presence of CP.
- Table 2 is not labeled
- In section 3.7 regarding pain management, the last sentence of the first paragraph states 35% of patients experience pain after surgery resulting in reduction of overall pain. Can you please elaborate on what this means?
- Did the authors consider reporting differences in opioid use in morphine milligram equivalents?
- In table 3 it says BMI is reported as both a median and a mean. It appears the mean is reported.
- In table 4, it appears that you are comparing PEI and PERT preoperatively vs postoperatively within DPPHR and PD. Would it not be expected that pancreatic function would decline postoperatively if a portion of the pancreas is resected? Furthermore, if you are comparing DPPHR vs PD, the statistical analysis should compare the incidence of new PEI, changes in PERT dosage, and new DM in DPPHR vs PD groups
Discussion
- It is difficult to come to the same conclusion regarding pain control as there are many factors that play a role in this. Given that the study spans a long time period and is non-randomized, there may be temporal confounding and selection bias at play that limit the ability to interpret the results.
- The conclusions regarding pain relief is not supported by the data (showing more opioid use and higher pain intensity in patients undergoing DPPHR). Additionally, there are significant limitations to the analysis including: lack of data about preoperative opioid use, missing data, and lack of adjustment for confounders.
- With regards to trends over time, could the authors comment on the generatability of the results?
Reviewer 2 Report
I read your paper and is very well written!
Maybe the numbers are not so impressive, but the quality of the writing is. In the discussion section maybe you can add a paragraph about malignancy as an indication - with less than 10% accuracy.
minor typo correction needed
Reviewer 3 Report
This is a retrospective observational study evaluating surgical outcomes for chronic pancreatitis with comparison between duodeno-preserving pancreatic head resection (DPPHR) and pancreatoduodenectomy (PD). The manuscript is well written and results are supported by adequate evidence.
It is e retrospective series with a limited number of patients in each group and a relatively short follow-up. All these factors are specified in the study limitations.
Comparison of the DPPHR and PD group found no difference in terms of morbidity, mortality, pain relief, and endocrine insufficiency. Patients in the DPPHR group had a significant lower rate of postoperative exocrine insufficiency. However, trends in pancreatic surgery over time are in favor of pancreatic resections probably because suspicion of malignancy is becoming the most common indication for surgery in specialized centers. I only suggest giving more emphasis to this issue, especially in the conclusions.
Round 2
Reviewer 1 Report
Thank you for considering my comments. The new version of the manuscript is much better. I recommend adding a brief sentence in the methods section to demonstrate that the authors were conscious of the missing data and chose to present it in the table for data transparency. For example, the authors can consider writing, "To increase the transparency of data quality and analysis, missing data will be presented in a separate column" or something like this.